# Inhibitory Effects and Mechanism of Action of Elsinochrome A on *Candida albicans* and Its Biofilm

**DOI:** 10.3390/jof8080841

**Published:** 2022-08-11

**Authors:** Lili Pan, Yuanyuan Yao, Hailin Zheng, Shuzhen Yan, Shuanglin Chen

**Affiliations:** 1Jiangsu Province Key Laboratory for Microbes and Functional Genomics, College of Life Sciences, Nanjing Normal University, Nanjing 210023, China; 2Jiangsu Province Key Laboratory of Molecular Biology of Dermatology and Venereal Diseases, Institute of Dermatology, Chinese Academy of Medical Sciences and Peking Union Medical College, Nanjing 210042, China

**Keywords:** *Candida albicans*, perylenequinonoid photosensitizers (PQP), Elsinochrome A, photodynamic antimicrobial chemotherapy, reactive oxygen species

## Abstract

Biofilm-associated *Candida albicans* infections, the leading cause of invasive candidiasis, can cause high mortality rates in immunocompromised patients. Photodynamic antimicrobial chemotherapy (PACT) is a promising approach for controlling infections caused by biofilm-associated *C. albicans*. This study shows the effect of Elsinochrome A (EA) against different stages of *C. albicans* biofilms in vitro by XTT reduction assay and crystal violet staining. The mechanism of action of EA on *C. albicans* biofilm was analyzed with flow cytometry, confocal laser microscopy, and the Real-Time Quantitative Reverse Transcription PCR (qRT-PCR). EA-mediated PACT significantly reduced the viability of *C. albicans*, with an inhibition rate on biofilm of 89.38% under a concentration of 32 μg/mL EA. We found that EA could not only inhibit the adhesion of *C. albicans* in the early stage of biofilm formation, but that it also had good effects on pre-formed mature biofilms with a clearance rate of 35.16%. It was observed that EA-mediated PACT promotes the production of a large amount of reactive oxygen species (ROS) in *C. albicans* and down-regulates the intracellular expression of oxidative-stress-related genes, which further disrupted the permeability of cell membranes, leading to mitochondrial and nuclear damage. These results indicate that EA has good photodynamic antagonizing activity against the *C. albicans* biofilm, and potential clinical value.

## 1. Introduction

*Candida albicans* is a common opportunistic pathogen. When the immune defense in the human body is disrupted, *C. albicans* transforms from a commensal organism into a pathogen, resulting in skin and mucous membrane infections or invasive fungal infections [1]. It can even lead to life-threatening systemic candidiasis in severe cases [2]. *Candida albicans* can form a protective dense biofilm [3], which acts as a barrier to prevent drug entry, as a mechanism of resistance against antifungal drugs [4]. With the growing problem of drug resistance in *C. albicans*, there is an urgent need to find new antifungal therapies and drugs that inhibit the formation and proliferation of biofilms caused by *C. albicans*.

Photodynamic antimicrobial chemotherapy (PACT) is one of the promising alternatives for traditional antibiotic fungal infection treatment, using photosensitizers, which are activated under the irradiation of a laser light with a specific wavelength to generate cytotoxic free radicals and oxidative substances that are harmful to cells, thereby damaging target cells at the site of action [5]. There have been many studies on the important role of photodynamic therapy using photosensitizers in the treatment of infectious diseases caused by various pathogens, including *C. albicans* [6,7,8]. The predominant strategy in this field is known as PACT [9]. In recent years, PACT has been widely used in the treatment of *C. albicans*, and breakthroughs have been made in inhibiting the formation of *C. albicans* biofilm [10,11,12,13]. The selection of photosensitizers is crucial for PACT. Studies have shown that hypocrellin, a perylenequinonoid photosensitizer (PQP), can effectively inhibit the formation and proliferation of *C. albicans* biofilm [14]. Elsinochrome A (EA), a fungal perylene quinone photosensitizer [15], was first reported by Chen et al., isolated from *Elsinoë* spp. I [16], and its structure was determined by Meille et al. [17]. Due to its excellent photosensitive properties, such as high quantum yields both of triplet and singlet oxygen, as well as a high photodynamic killing effect on cancer cells [18,19,20] causing the oxidation and damage of target cells [21], it is currently known as an excellent photosensitizer in the visible light region and is expected to be developed as a new clinically phototherapeutic medicine. However, the antifungal effect of EA on *C. albicans* has never been studied. This study explored the effect of EA on the formation of the *C. albicans* biofilm and the associated mechanism of action, aiming to lay a theoretical foundation for the application of EA-mediated PACT on treating candidiasis caused by *C. albicans*.

## 2. Materials and Methods

### 2.1. Strains and Growth Conditions 

The *C. albicans* strain CAMS-CCPM-D-52061 was collected in Qilu Hospital of Shandong University and was isolated from vaginal secretions of a patient on 1 March 2009. It was preserved at the Department of Medical Mycology, Institute of Dermatology, Chinese Academy of Medical Science and Peking Union Medical College. Before each experiment, the isolate was cultured in a liquid YPD (Yeast Peptone Dextrose Agar) medium overnight at 37 °C and 225 rpm. Elsinochrome A (98%) was previously isolated from the strain S4201 (CGMCC No10816) of *Shiraia bambusicola* in our lab. Firstly, the strain S4201 was fermented for 16 d in rice medium (100 g rice, 100 mL water) at 28 °C. The solid-state fermentation material was heat-dried and crushed. Then, the extraction was carried out using the polar gradient extraction method of petroleum ether–ethyl acetate–ethanol, and the extract was purified by column chromatography separation and purification. Elsinochrome A crystals were obtained after elution and volatilization with ethyl acetate–petroleum ether [22]. Elsinochrome A is insoluble in water and soluble in organic solvents, so DMSO was used as the solvent in this study. A stock solution of EA dissolved in DMSO was prepared and stored at −4 °C in the dark. Fluconazole (FLC) (Solarbio, Beijing, China), was used as a positive control. The irradiation power was a light emitting diode (LED) lamp (90 mW/cm^2^, Cidly Optoelectronic Technology Co., Ltd., Shenzhen, China)with a wavelength at 470 nm.

### 2.2. Antifungal Susceptibility Testing

To determine the antifungal activity of EA against the *C. albicans* strain CAMS-CCPM-D-52061, the minimum inhibitory concentration (MIC) was determined by the broth microdilution assay under the recommended guidelines of the Clinical and Laboratory Standards Institute (CLSI), reference document M27-A3 [23], with some modifications. Briefly, *C. albicans* suspensions (adjusted to 1 × 10^6^ CFU/mL in RPMI 1640 medium) were inoculated in a 96-well plate and incubated at room temperature with different EA solutions (in a concentration gradient series as 0.0625, 0.125, 0.25, 0.5, 1, 2, 4, 8, 16, 32, 64, 128, 256, 512, and 1024 μg/mL) made up to the final volume of 200 μL/well. Cells incubated with FLC solution (the same concentration range as EA) alone were included as a control and without EA as blank. Then, the plates were immediately treated by light irradiation (wavelength range: 470 nm; power: 90 mW/cm^2^; irradiation distance: 10 cm; energy density: 324 J/cm^2^). After irradiation for 60 min, the MIC endpoint was determined after 24 h of dark incubation at 37 °C, in comparison with the EA-free growth control wells. The MIC value was defined as the lowest concentration of EA that completely inhibited the growth of *C. albicans*. 

### 2.3. Effect of PACT on the Growth Kinetics of C. albicans

*Candida albicans* (10^6^ CFU/mL) suspensions were treated with EA (2, 4, 8, 16, and 32 μg/mL), while control samples were incubated with RPMI 1640 medium only. After irradiation treatment for 60 min and being cultured in the dark at 37 °C, the contents of the wells at 0, 6, 12, 24, and 48 h of each culture were taken for gradient dilution. A total of 50 μL of each dilution was plated onto Sabouraud Dextrose Agar (SDA) plates and incubated in the dark at 37 °C for 24 h. The CFUs were then counted. A time–inhibition curve was made based on the logarithmic values of cell numbers (log CFU/mL) in each culture.

### 2.4. Effect of PACT on Biofilm Formation

According to reference [24], the antifungal activity of EA against *C. albicans* biofilm was determined by the reduction of XTT (2,3-bis(2-methoxy-4-nitro-5-sulfophenyl)-2H-tetrazolium-5-carboxanilide sodium salt) (Sigma, St. Louis, MO, USA) assay. A XTT solution (0.5 g/L) was mixed with 10 mM menadione solution (Solarbio, Beijing, China) at a final concentration of menadione of 1μM. The *C. albicans* suspensions (10^6^ CFU/mL) were inoculated with 100 μL of different EA solutions (in a concentration gradient series as 4, 8, 16, 32, 64, 128, 256, 512, and 1024 μg/mL), to the final volume of 200 μL/well. Cells incubated in FLC solution alone were included as a control and without EA as a blank. The suspensions were then illuminated (470 nm, 90 mW/cm^2^, 60 min). After irradiation, the contents of the wells remained at rest for 24 h in the dark at 37 °C until sampling. After incubation, the liquid in the samples was aspirated. Each well was washed twice with PBS to remove the non-adherent cells. Then, XTT–menaquinone solution was added to the plates, which were then incubated in the dark for 2 h. After incubation, the light absorbance of the reduced formazan-colored product was measured at 490 nm (OD490) in a SpectraMax M2 Microplate reader. Compared with the blank group, the drug concentration of the group with an OD value reduced by 80% was defined as SMIC_80_.

### 2.5. Time-Dependent Photoinactivation Effect

The *C. albicans* suspensions (10^6^ CFU/mL) were mixed with 2, 4, 8, 16, and 32 μg/mL concentrations of EA in equal volume. The culture was then immediately illuminated (470 nm, 90 mW/cm^2^) for 5, 10, 20, 40, 60, 80 and 100 min, respectively. After irradiation, the cells were incubated at 37 °C in the dark for 24 h. Following incubation, the biofilm was assessed using XTT assay as previously described.

### 2.6. Biofilm Initial Cell Attachment Assay

#### 2.6.1. XTT Assay for the Effect of EA on the Adhesion of *C. albicans*

The XTT assay was used to determine the inhibitory effect of EA-PACT on the attachment of *C. albicans* according to the method described in reference [25] and with some modifications. An equal volume of different concentrations of EA solution (2, 4, 8, 16, and 32 μg/mL) was added to the *C. albicans* suspensions (10^6^ CFU/mL) on 96-well titer plates, mixed well. Each plate was then statically incubated for 90 min at 37 °C with irradiation (470 nm, 90 mW/cm^2^, 60 min). Subsequently, plates were washed twice with PBS buffer, and XTT–menaquinone solution was added to each well. Plates were then incubated in the dark for 2 h at 37 °C. The absorbance at 490 nm was measured using a microtitre plate reader.

#### 2.6.2. Adhesion-Related Gene Transcription Analysis by qRT-PCR Assays

An equal volume of different concentrations of EA solution (2, 4, 8, 16, and 32 μg/mL) was added to the *C. albicans* suspensions (10^6^ CFU/mL) on a 96-well titer plate, mixed well. Each plate was then statically incubated for 90 min at 37 °C with irradiation (470 nm, 90 mW/cm^2^, 60 min). After incubation, the cells were collected and washed by centrifugation at 1500× *g* for 3 min at 4 °C. The total RNA was isolated by using TRIzol^®^ reagent. As previously described [26], cDNA was synthesized using the HiScript^®^ III RT SuperMix kit (Vazyme, Nanjing, China) according to the manufacturer’s instructions. Real-time qPCR was performed using a SYBR green master mix in an ABI StepOne Real-Time PCR System (ABI, ABI StepOnePlus, Foster City, CA, USA). Using the housekeeping gene *ACT1* as an internal control, the formula 2^−ΔΔCT^ was used to quantify the relative mRNA levels of *C. albicans* biofilm adhesion-related genes (*MP65* and *ALS3* genes) with primer sequences (Table 1).

### 2.7. Biofilm Eradication Assay

For pre-formed 24-h *C. albicans* biofilm, *C. albicans* suspensions (10^6^ CFU/mL) were transferred to each well of the microtitre plate and incubated statically for 24 h at 37 °C. After incubation, plates were washed twice with PBS, and equal volumes of EA solutions with different concentrations (2, 4, 8, 16, 32, 64, 128, 256, 512, and 1024 μg/mL) were added. Then, plates were irradiated (470 nm, 90 mW/cm^2^, 60 min), with the wells without EA acting as blanks for the subsequent steps. Following irradiation, the biofilm was assessed using XTT assay, as previously described, and crystal violet (CV) staining assay. For CV methodology, cells were fixed with methanol for 15 min and stained with 1% CV solution for 15 min. Next, the plates were washed with sterile water and absolute ethanol was added to release the dye from the biofilm. After 30 min, the absorbance was measured at 570 nm using a microtitre plate reader.

For pre-formed 48-h *C. albicans* biofilm, biofilms were prepared as described above. After static incubation at 37 °C for 48 h, plates were washed twice with PBS, and various drug concentrations were added to each well to a final volume 100 µL. After irradiation for 60 min, plates were washed with PBS and assessed using the XTT assay and CV staining as described above. 

### 2.8. Effect of PACT on Cell Permeability

*Candida albicans* cells (10^6^ CFU/mL) were treated with EA solutions (8, 16, and 32 μg/mL), and then incubated in a 37 °C incubator for 6 h under dark conditions after irradiation (470 nm, 90 mW/cm^2^, 60 min). As previously reported [27], first, the culture was washed twice with PBS buffer and PI (propidium iodide) stained (30 μM) for 30 min at 37 °C in the dark. After PBS washing, the slices were made and observed under a fluorescence microscope (DMI6000B, Leica, Wetzlar, Germany).

### 2.9. Effect of PACT on Relative Electric Conductivity of Cell Membrane

The concentration of the *C. albicans* suspensions was adjusted to 10^6^ CFU/mL. As previously described [28], *C. albicans* suspensions were taken in a water bath (100 °C, 10 min). After cooling, they were used as a positive control (G0). Then, *C. albicans* suspensions were mixed with different concentrations of EA solutions—2, 4, 8, 16, 32, and 64 μg/mL, respectively. After irradiation (470 nm, 90 mW/cm^2^, 60 min), the supernatant was centrifuged for measuring the conductivity with a conductivity meter (DDS-11A, Shanghai Precision Scientific Instrument Co., Ltd., Shanghai, China), recorded as G1. The formula for relative conductivity is
relative conductivity (%) = (G1/G0) × 100

### 2.10. Effect of PACT on Plasma Membrane Dynamics

*Candida albicans* cells (10^6^ CFU/mL) were treated with EA solutions (8, 16, and 32 μg/mL) for 2 h (undergoing irradiation for 60 min) at 37 °C, and the treatment group without EA was used as a control. Then, the cells were fixed with 0.37% formaldehyde for 30 min, washed twice with PBS, and snap-frozen in liquid nitrogen. After thawing, the cells were resuspended and incubated with 0.6 mM 1,6-diphenyl-1,3,5-hexatriene (DPH) for 45 min. Next, the stained cells were measured by a fluorescence microplate reader to detect the absorbance values of excitation light at 350 nm and absorption light at 425 nm, and the relative fluorescence intensity was calculated according to the absorbance values of each concentration [29].

### 2.11. Effect of PACT on ROS Production

#### 2.11.1. Determination of ROS Production

*Candida albicans* cells (10^6^ CFU/mL) were treated with EA (8, 16, and 32 μg/mL) after irradiation (470 nm, 90 mW/cm^2^, 60 min), and then incubated under 37 °C for 4 h under dark conditions. Following staining with 40 μM DCFH-DA (2′,7′-dichlorodihydrofluorescein diacetate) (St. Louis, MO, USA) for 30 min in the dark, the fluorescence intensity was measured after cell collection using a fluorescence microplate reader following the kit instructions. In addition, the stained cells were visualized by confocal laser scanning microscopy (CLSM) (Nikon, Ti-E-A1R, Tokyo, Japan) using a 63× objective lens.

#### 2.11.2. Oxidative-Stress-Related Genes Transcription Analysis by qRT-PCR Assays

The expression of *C. albicans* biofilm oxidative-stress-related genes (*CAP1*, *CAT1*, and *SOD1* genes) at the mRNA level was assessed by qRT-PCR as previously reported [26].

### 2.12. Mitochondrial Membrane Potential

The *C. albicans* suspensions (10^6^ CFU/mL) were treated with EA solutions at different concentrations (8, 16, and 32 μg/mL). After irradiation (470 nm, 90 mW/cm^2^, 60 min), cultures were incubated in the dark at 37 °C for 2 h, and the treatment without EA was used as the control group. Then, the samples were stained with JC-1 working solution for 20 min in dark as the instructions of the JC-1 mitochondrial membrane potential assay kit (San Jose, CA, USA). The cells with different treatments were then washed and detected by flow cytometry (BD, FACSverse, Franklin Lakes, NJ, USA).

### 2.13. Detection of DNA Fragmentation

*Candida albicans* cells (10^6^ CFU/mL) were treated with EA solutions (8, 16, and 32 μg/mL), after irradiation (470 nm, 90 mW/cm^2^, 60 min), and then incubated at 37 °C for 6 h under dark conditions, and the treatment group without EA was used as a control. The cells were fixed with 70% ethanol for 10 min at room temperature, washed with PBS, and stained with 10 μg/mL DAPI (diamidino-phenyl-indole) (Solarbio, Beijing, China) for 30 min in the dark. Then, the cells were mounted onto glass slides and observed by fluorescence microscopy (DMI6000B, Leica, Wetzlar, Germany), as described previously [30].

**Table 1 jof-08-00841-t001:** Gene-specific primers used for the relative quantification of gene expression by RT-PCR.

Primers	Sequence	References
*MP65*-F	TCAACACTGAACCACCTC	[31]
*MP65*-R	ATACCTTTAGCACCACCA	[31]
*ALS1*-F	CATCATTGACTCAGTTGT	[32]
*ALS1*-R	CAGTGGAAGTAGATTGTG	[32]
*CAP1*-F	AGTCAATTCAATGTTCAAG	[32]
*CAP1*-R	AATGGTAATGTCCTCAAG	[32]
*CAT1*-F	GACTGCTTACATTCAAAC	[32]
*CAT1*-R	AACTTACCAAATCTTCTCA	[32]
*SOD1*-F	TTGAACAAGAATCCGAATCC	[33]
*SOD1*-R	AGCCAATGACACCACAAGCAG	[33]
*ACT1*-F	TTGATTTGGCTGGTAGAG	[34]
*ACT1*-R	ATGGCAGAAGATTGAGAA	[34]

### 2.14. Statistical Analysis 

Prism 8.0 software (GraphPad Software, San Diego, CA, USA) was used for statistical analysis. All results were expressed as the mean ± SD. One-way analysis of variance (ANOVA) was used for comparisons of the means. *p* values less than 0.05 were considered statistically significant.

## 3. Results

### 3.1. EA-Mediated PACT Decreases C. albicans Survival

To evaluate the PACT efficacy of the EA, we determined the MIC against *C. albicans* (Table 2). Elsinochrome A exhibited powerful antifungal activity after irradiation. The MIC value of EA was 1 μg/mL. The time–inhibition curve assay was also performed to evaluate the antifungal activity of EA at different concentrations (Figure 1). The results confirm that the EA was sufficient to rapidly kill *C. albicans* after irradiation. The EA treatment groups of each concentration had obvious antifungal effects. The yeast viability decreased by 3.09 log CFU mL^−1^ and 3.37 log CFU mL^−1^ at the treatment concentrations 16 μg/mL and 32 μg/mL, respectively. These results imply that EA has strong antifungal activity against *C. albicans*. After culturing for 24 h, the cell viability in each EA treatment group tended to be stable.

Sessile minimum inhibitory concentration (SMIC) was determined for the *C. albicans*: SMIC_80_ was the anticandidal concentration at which an 80% decrease in biofilm OD490 was detected in comparison with the control biofilms formed by the same *C. albicans* isolate in the absence of anticandidal agent. The minimum inhibitory concentration (MIC) was defined as the lowest concentration of EA that completely inhibited the growth of *C. albicans*.

### 3.2. EA-Mediated PACT Inhibits C. albicans Biofilm Formation

To provide a more detailed quantitative assessment of the inhibitory effect of EA on *C. albicans* biofilms, an XTT reduction assay was performed to measure the SMIC_80_. The results show that the SMIC_80_ of EA against *C. albicans* biofilm was 16 μg/mL (Table 2), while the SMIC_80_ of antifungal drug FLC against *C. albicans* biofilm was greater than 1024 μg/mL. It can be seen that EA has strong anti-*C. albicans* activity. Additionally, compared with the traditional antifungal drug FLC, EA-mediated PACT shows stronger inhibitory activity against *C. albicans* biofilm.

For PACT, the effect of irradiation time on photosensitization and *C. albicans* biofilm formation is very important. Therefore, we investigated the time-dependent inhibition rate of EA-mediated PACT on biofilm formation by XTT reduction assay. The PACT effect of EA on *C. albicans* biofilm was time- and concentration-dependent (Figure 2). The inhibitory effect of EA on biofilm formation was also enhanced with the increase in administration concentration. When *C. albicans* was treated with 32 μg/mL EA and irradiated for 60 min, immediately after incubating for 24 h, the inhibition rate of EA on biofilm formation was as high as 83.92%, which was significantly higher than that of the control group (*p* < 0.05). In addition, with the increase in irradiation time, the inhibition rate gradually increased. When the irradiation time reached 60 min, the inhibitory rate of EA on *C. albicans* biofilm reached the highest (31.82%, 52.45%, 69.08%, 83.41%, and 89.38% for 2, 4, 8, 16, and 32 μg/mL EA, respectively). Therefore, 60 min was selected as the treatment time of illumination in the subsequent experiments.

### 3.3. EA-Mediated PACT Has Anti-Biofilm Activity toward Two Phases of C. albicans Biofilm Maturation

In the cell adhesion stage of *C. albicans* (when it was inoculated into a 96-well plate and treated for 90 min), the results showed that the treatment with 32 μg/mL EA could significantly inhibit the metabolic activity of *C. albicans* cells, with an inhibition rate as high as 72.71% (Figure 3A). We then quantified the expression of *C. albicans* biofilm adhesion-related genes (*MP65* and *ALS3* genes), which play an important role in mediating the adhesion process of *C. albicans* by qRT-PCR. The results show that the *C. albicans* biofilm adhesion-related genes *MP65* and *ALS3* genes were down-regulated to varying degrees by EA treatment (Figure 3B). When the administration concentration of EA was 32 μg/mL, the expression levels of *MP65* and *ALS31* genes decreased by 40.79% and 63.58%, respectively (*p* < 0.05).

Next, we evaluated the scavenging effect of EA on pre-formed biofilms. For pre-formed *C. albicans* in the form of 24 h early biofilm and 48 h mature biofilm, the CV methodology was used to measure the total biofilm mass. Colorimetric dye XTT was used to measure the total metabolic activity of *C. albicans*. We found that the removal activity of EA on 48 h mature biofilms was as significant as that on 24 h early biofilms (Figure 4). Pre-formed fungal biofilm of *C. albicans* was dose-dependently affected by EA. The detection of changes in total biofilm showed that the 24 h early biofilm and 48 h mature biofilm of *C. albicans* that had formed began to be eradicated after treatment with EA and irradiation, and the highest eradication rates reached 39.49% and 35.16%, respectively. The metabolic activity of *C. albicans* early biofilm (24 h) and mature biofilm (48 h) was reduced by 39.71% and 38.53%, respectively.

### 3.4. Effect of EA-Mediated PACT on Cell Membrane

#### 3.4.1. Permeability of Cell Membrane

In the cell membrane, the change in the relative electric conductivity is often used to indicate severe and irreversible damage to the cytoplasm and plasma membrane [35]. Changes in relative electric conductivity can reflect the impact of EA on cell membrane permeability. Our results show that after irradiation, the relative electric conductivity of the *C. albicans* cell membrane at each EA concentration was significantly higher than without EA (Figure 5A, *p* < 0.05), and the effect was dose dependent. When the dose was increased to 64μg/mL, the relative electric conductivity of the *C. albicans* cell solution was 68.36%, 1.85 times that of the control group. These results confirm that EA-mediated PACT damaged the cell membrane of *C. albicans* and increased its cell membrane permeability.

#### 3.4.2. Integrity of Cell Membrane

The effects of EA-mediated PACT on the cell membrane integrity of *C. albicans* were detected by the fluorescent dyes DPH and PI, respectively. Propidium iodide binds to nucleic acids and emits red fluorescence, which only penetrates the cells if the cell membrane is damaged. In this study, fluorescence microscopy showed that few cells in the control group were damaged, while many cells in the EA treatment group were. The proportion of *C. albicans* cells stained by PI increased with increasing EA concentration (Figure 6), indicating that EA treatment significantly increased the number of PI-stained cells. When the concentration of EA reached 32 μg/mL, the number of *C. albicans* cells was significantly reduced, and the cell membrane was damaged. 

The dye DPH can bind to the hydrocarbon tail of phospholipids without affecting the bilayer structure of phospholipids [36]. The fluorescence intensity of DPH indicated that EA attenuated the fluorescence signal intensity of DPH in a dose-dependent manner. When the cells were treated with 8, 16, and 32 μg/mL EA, the DPH fluorescence intensity decreased to 78.32%, 67.16%, and 38.88%, respectively (Figure 5B). The fluorescence anisotropy of DPH on the plasma membrane decreased significantly with the increase in EA concentration, which reflects the antifungal activity of EA-mediated PACT on *C. albicans* cells by perturbing the plasma membrane. Thus, it was confirmed that EA-mediated PACT disrupted the integrity of the cell membrane and increased the permeability of the cell membrane

### 3.5. EA-Mediated PACT Induces ROS Production

As depicted in Figure 7, a significant increase was observed in the fluorescence intensity of *C. albicans* cells after PACT by confocal laser scanning microscopy, which represented a significant increase in ROS in a dose-dependent manner after treatment with EA. Compared to the control cells, the ROS level increased by 3.14, 4.92, and 13.49 times when treated with 8, 16, and 32 μg/mL EA, respectively, which was in accordance with the observation by laser confocal microscopy (Figure 8A). In addition, the qRT-PCR results show that EA inhibited the expression of the oxidative-stress-related genes *CAP1*, *CAT1*, and *SOD1* in *C. albicans* cells to varying degrees. When treated with 32 μg/mL EA, the expression of *CAP1*, *CAT1*, and *SOD1* genes decreased by 36.63%, 72.66%, and 62.57%, respectively (Figure 8B). It has been shown that EA induces a mass amount of ROS production in *C. albicans* cells and down-regulates the expression of oxidative-stress-related genes, resulting in cell oxidative damage and membrane damage. 

### 3.6. EA-Mediated PACT Decreases Mitochondrial Membrane Potential and Induces Nuclear Fragmentation

The mitochondrial membrane potential (MMP) is an important hallmark of mitochondrial function, and dissipation in MMP is a sign of early apoptosis [37]. Therefore, we measured the relative value of mitochondrial transmembrane potential upon treatment with EA. The upper-right quadrant represents the relative value of mitochondrial membrane potential. Flow cytometry shows that the relative value of the mitochondrial membrane potential of *C. albicans* cells decreased, and the proportion of the decrease in cells’ mitochondrial membrane potential increased with the increase in EA concentration. Treatment with 8, 16, and 32 μg/mL EA resulted in a decrease in the relative value of mitochondrial membrane potential in *C. albicans* cells by 7.87%, 9.16%, and 24.3%, respectively (Figure 9). The results suggest that EA could induce mitochondrial damage in *C. albicans* cells after irradiation.

Nuclear fragmentation is one of the well-recognized characteristics of late apoptosis. We measured nuclear fragmentation with DAPI staining as an indicator [38]. Fluorescence microscopy analysis revealed that the appearance of *C. albicans* cells in the EA treatment group showed chromatin debris distributed in the cells (Figure 10A), while the control group was normal, showing bright round nuclei (Figure 10B), indicating that the cells developed apoptotic characteristics after the treatment with EA, resulting in nuclear damage. 

## 4. Discussion

### 4.1. EA-Mediated PACT Effect on C. albicans

*Candida albicans* can cause severe systemic and mucosal infections in immunocompromised people [39,40]. Mature biofilms exhibit stronger resistance to many antifungal drugs, making the treatment of these infections particularly difficult [41]. Microbes embedded in biofilms are 100–1000 times more resistant to conventional antibiotic drugs than planktonic bacterial cells [42]. Therefore, it is of great significance to develop antifungal therapies that effectively remove biofilms. 

Photodynamic antimicrobial chemotherapy utilizes photosensitizers accumulated in cells to generate ROS in the presence of ambient oxygen through specific visible light irradiation, which attacks intracellular macromolecules and the biofilm matrix, resulting in cell damage and even death [43]. Compared with traditional antibiotic therapy, PACT has many advantages, including fewer side effects, higher safety, and an efficacy unaffected by common microbial resistance [44]. Studies have shown that photosensitizer-mediated PACT, such as methylene blue (MB) [10], toluidine blue O (TBO) [11], photofrin [12], and benzo[a]phenoxazinium [13], can effectively inhibit or destroy *C. albicans* biofilms. However, as a photosensitizer with excellent performance among perylene quinone compounds, the antifungal effect of EA on *C. albicans* and the effectiveness of destroying or removing biofilm are still unclear. This study was the first to elucidate the effect of EA on *C. albicans* and its biofilm under light conditions. The results show that EA has value in the PACT of *C. albicans* infection and can significantly inhibit the growth of *C. albicans* planktonic cells and biofilm. Moreover, the inhibitory effect of EA (MIC = 1 μg/mL, SMIC_80_ = 16 μg/mL) against *C. albicans* in this study is stronger than that of the traditional antifungal drug FLC (MIC = 0.5 μg/mL, SMIC_80_ > 1024), which is one of the most frequently used antifungal drugs for the prevention and treatment of *C. albicans* infections [45]. 

### 4.2. EA-Mediated PACT Effect on C. albicans Biofilm

Biofilm formation is a critical stage in *C. albicans* colonization and infection [43]. The formation of *C. albicans* biofilm can be divided into three stages: early cell adhesion (0–11 h), mid-term development stage (12–30 h), and late-stage mature biofilm formation (30–72 h) [46]. Thus, the inhibition of naive cell adhesion and the clearance of mature biofilms are beneficial for the treatment of invasive candidiasis. Previous studies have shown that proteins encoded by the *ALS3* and *MP65* genes contribute to the adhesion of *C. albicans* [26]. The results of this study showed that the treatment with 32 μg/mL EA could significantly inhibit the expression of biofilm adhesion-related genes under light conditions, with an inhibition rate of 71.37% for *C. albicans* in the adhesion stage. In different stages of *C. albicans* biofilm formation, the inhibition rate of EA was also distinct: when *C. albicans* was incubated with 32 μg/mL EA for 24 h, the inhibition rate of EA on *C. albicans* biofilm was 89.38%; for the pre-formed *C. albicans* biofilms matured at 24 h and 48 h, the inhibition rates of EA reached 35.3% and 38.38%, respectively. Although the rate of inhibition of pre-formed mature biofilms by EA is lower than that measured after the simultaneous incubation of *C. albicans* and EA, the former is more clinically relevant because biofilm formation tends to precede treatment, which contributes to stronger drug resistance [42]. In this study, EA-mediated PACT not only exhibited significant scavenging effects on pre-formed biofilms at different stages, but also effectively reduced the cell viability and metabolic activity of *C. albicans* in biofilms, indicating that it has the potential to reduce cell colonization.

### 4.3. Mechanism of Action of EA-Mediated PACT

Maintaining the integrity of the cytoplasmic membrane is critical for many essential functions of microbial pathogens [47]. To understand whether EA exerts its antifungal effect by targeting the cell membrane, we investigated the relevant effect by staining with the fluorescent dyes PI and DPH to analyze the cell membrane dynamics and permeability, respectively, and measuring the relative conductivity of the cell membrane of *C. albicans*. The dye PI can penetrate damaged or permeable cell membranes and fluoresces red, while DPH readily binds to the hydrocarbon tail region of phospholipids to detect changes in *C. albicans* cell membrane dynamics [48]. Treatment with EA remarkably increased the number of PI-stained cells and the relative conductivity of the *C. albicans* plasma membrane, and decreased the DPH fluorescence intensity, implying that EA-mediated PACT damaged the integrity of the cell membrane, which affected the normal physiological function of the plasma membrane. 

Reactive oxygen species (ROS) play a key role in normal cellular metabolism. However, excessive ROS damages not only various macromolecules in cells, such as nucleic acids, lipids, and proteins, but also cell membrane permeability, which plays an important role in mediating aspects of programmed cell death, such as apoptosis [49,50]. To determine the production and accumulation of ROS in response to EA treatment in *C. albicans* cells, we used the fluorescent dye DCFH-DA in this study to detect the level of intracellular ROS production [51]. The results confirm that the intracellular ROS level of *C. albicans* after EA treatment was significantly increased in a dose-dependent manner, indicating that the high-level accumulation of intracellular ROS content belongs to a part of the mechanism by which EA mediates PACT. A high ROS level is a hallmark of mitochondrial dysfunction in yeast cells [52]. In this study, mitochondrial dysfunction was assessed by detecting the decrease in mitochondrial membrane potential. The results show that the mitochondrial membrane potential of *C. albicans* cells significantly decreased after EA treatment, suggesting that mitochondrial function was damaged. Decreased mitochondrial membrane potential is a characteristic event in the early stages of apoptosis. In addition, it was also observed that the nucleus of *C. albicans* was damaged, and morphological changes with characteristics of apoptosis, such as chromatin fragments, appeared. Above all, we speculate that the high-level accumulation of ROS is important in the photodynamic sterilization mechanism of EA against *C. albicans*. Cells exposed to EA under specific wavelengths of light produce large amounts of ROS that directly attack a variety of biological macromolecules, including biofilm substrates (such as polysaccharides), cell surfaces (such as lipids), and intracellular molecules (such as proteins and DNA), causing multi-target oxidative damage. The ROS directly damages the cell membrane, resulting in a significant increase in cell membrane permeability, prompting more EA to enter the cell. Excessive ROS destroys the antioxidant defense mechanism in microbial cells, leading to *C. albicans* cell metabolism disorder, mitochondrial dysfunction, nuclear damage, and eventually cell death, which collapses the biofilm matrix and disintegrates microbial cells. 

But there is still a long way to go before putting EA into clinical use. First, as the results were obtained only from one strain of *C. albicans*, we still lack data on other strains which are resistant to antifungals with different mechanisms, like mutations in genes associated with efflux pumps. Second, further in vivo investigations are important to more deeply explore its clinical values. A further in-depth study of the specific pathways and mechanisms of EA in vivo against *C. albicans*, and experiments determining the safety, e.g., regarding the toxicity for human cells, will help to realize the clinical application of EA-mediated PACT for *C. albicans* infection. Moreover, the hydrophobic structure of EA limits its application in PACT. Targeted drug delivery systems can not only solve the water solubility problem of photosensitizers, but also increase the accumulation of drugs in target tissues and enhance their efficacy [53]. Thus, preparing a nano-targeted drug delivery system with high water solubility will be one of the most important methods of improving the EA delivery efficiency and therapeutic effect, which will provide a new therapeutic strategy for the treatment of invasive fungal infections and advance the clinical application of PACT in the treatment of candidiasis.

## 5. Conclusions

In sum, this study is the first to explore the antifungal effect of EA-mediated PACT on *C. albicans* biofilms in vitro. The treatment with 1024 μg/mL Elsinochrome A and irradiation (470 nm; 90 mW/cm^2^; 324 J/cm^2^) had significant effects on pre-formed mature biofilms, with a clearance rate of 35.16%. After irradiation, EA destroys the cell membrane by generating a large amount of ROS, damages the normal function of the mitochondria, and eventually leads to apoptosis, suggesting that EA has a greater potential for clinical application as an antifungal agent and for the treatment of candidiasis.

## Figures and Tables

**Figure 1 jof-08-00841-f001:**
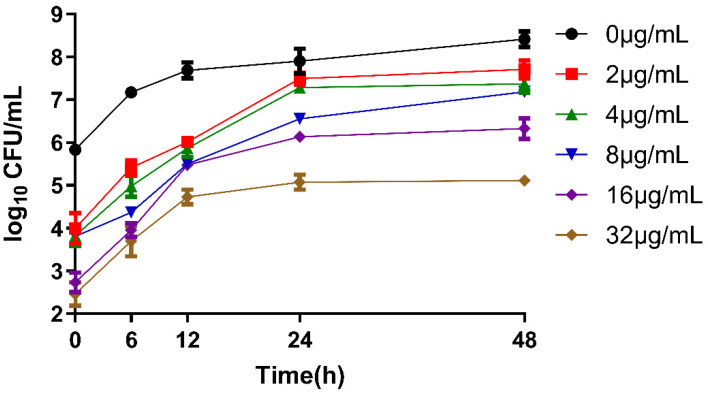
Time–kill curves of EA against *C. albicans*.

**Figure 2 jof-08-00841-f002:**
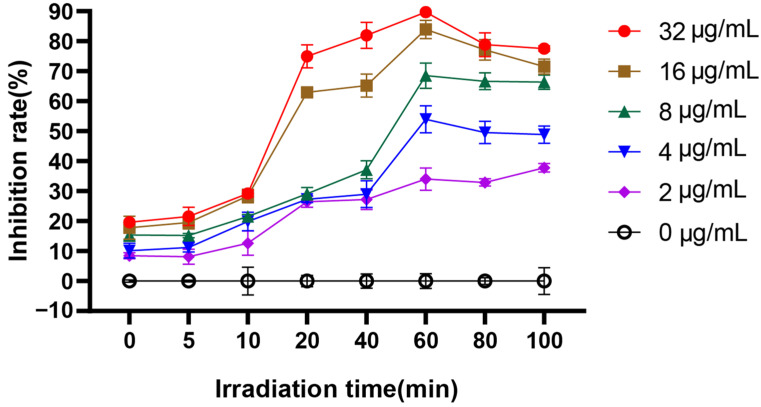
The effect of irradiation time on the inhibition rate of *C. albicans* biofilm.

**Figure 3 jof-08-00841-f003:**
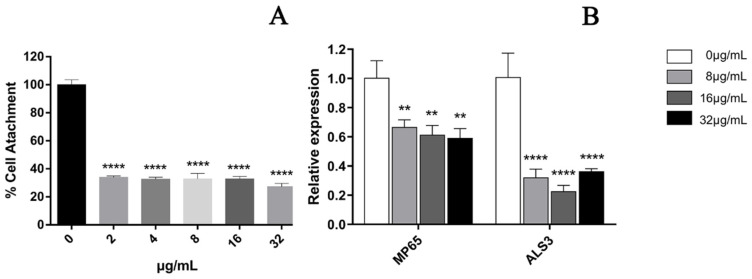
Effect of EA on adhesion of *C. albicans*. (**A**) XTT assay was used to detect the effects of EA on the metabolic activity of *C. albicans* during the adhesion. (**B**) qRT-PCR was used to detect the effects of EA on the expression of adhesion-related genes in *C. albicans*. Values are expressed as mean ± SD; *n* = 3; ** *p* < 0.01, and **** *p* < 0.0001 when compared to the control.

**Figure 4 jof-08-00841-f004:**
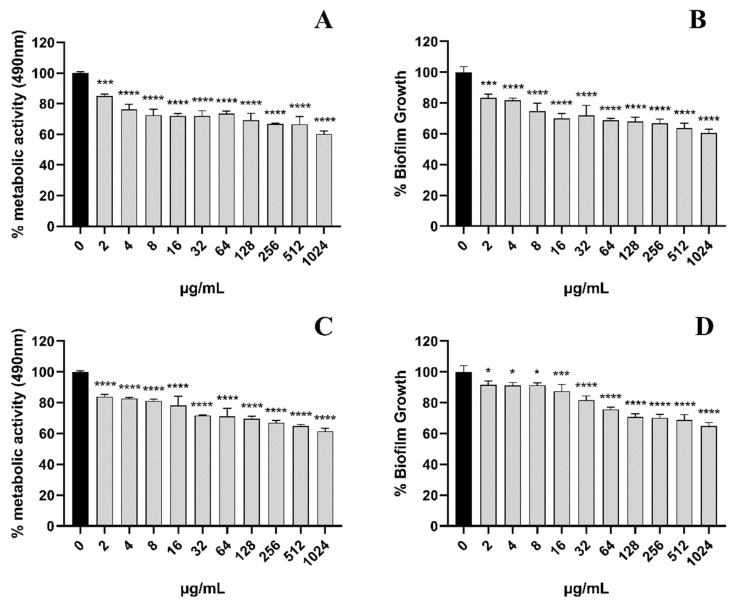
The scavenging effect of EA on the *C. albicans* 24-h early biofilm and 48-h mature biofilm. The clearance of 24-h early biofilm formation of *C. albicans* by various concentrations of EA as measured by XTT dye (**A**) and 1% CV (**B**). The 48-h mature biofilm of *C. albicans* eradication as measured by XTT dye (**C**) and 1% CV (**D**). Values are expressed as mean SD; *n* = 3; * *p* < 0.05, *** *p* < 0.001, and **** *p* < 0.0001 when compared to the control.

**Figure 5 jof-08-00841-f005:**
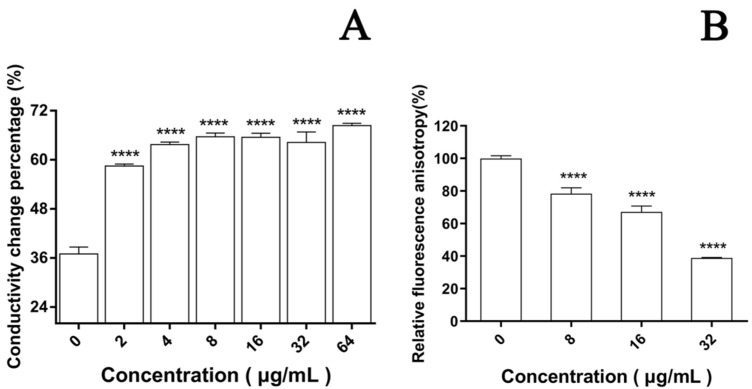
Effect of EA on cell membrane. (**A**) The effect of EA on the relative conductivity of the cell membrane of *C. albicans*. (**B**) The effect of EA on the cell membrane dynamic. Values are expressed as mean ± SD; *n* = 3; **** *p* < 0.0001 when compared to the control.

**Figure 6 jof-08-00841-f006:**
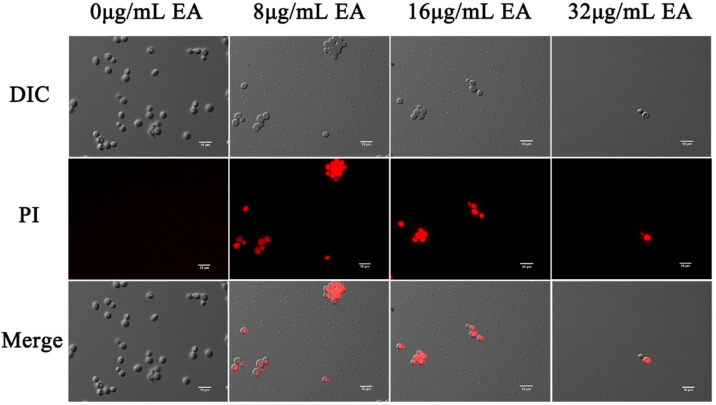
Effect of EA on the cell membrane permeability of *C. albicans*. Cells were observed under a Nikon 80I fluorescence microscope (63× oil immersion objective). Scale bars represent 10 μm.

**Figure 7 jof-08-00841-f007:**
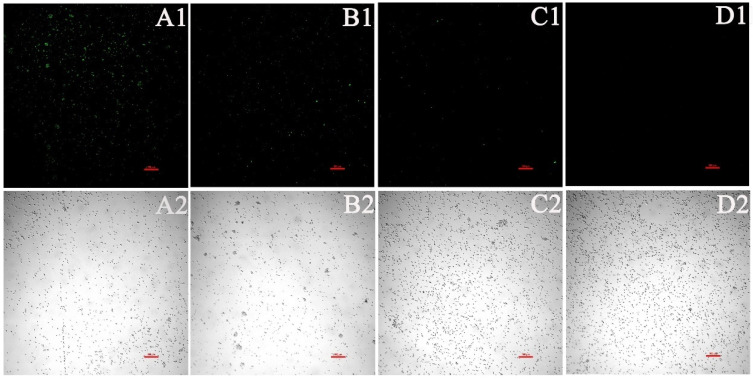
Detection of ROS level in *C. albicans* by CLSM. (**A1**,**A2**) were treated with 32 μg/mL EA; (**B1**,**B2**) were treated with 16 μg/mL EA; (**C1**,**C2**) were treated with 8 μg/mL EA; and (**D1**,**D2**) were the untreated controls. Scale bars represent 100 μm.

**Figure 8 jof-08-00841-f008:**
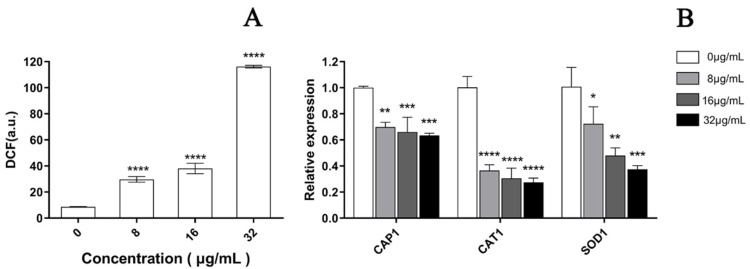
Effect of EA on the level of intracellular ROS of *C. albicans*. (**A**) The ROS quantitative results. Intracellular ROS can oxidize non-fluorescent DCFH to generate fluorescent DCF. Detection of DCF fluorescence can know the level of intracellular ROS. (**B**) The effect of EA on the expression of oxidative-stress-related genes in *C. albicans*. Values are expressed as mean ± SD; *n* = 3; * *p* < 0.05, ** *p* < 0.01, *** *p* < 0.001, and **** *p* < 0.0001 when compared to the corresponding controls.

**Figure 9 jof-08-00841-f009:**
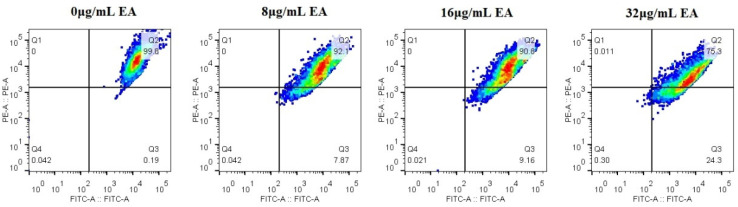
Effect of EA on the mitochondrial membrane potential of *C. albicans*. The JC-1 staining was used to detect the mitochondrial membrane potential ΔΨm. Double positive for FL1 (FITC channel) and FL2 (PE channel) on flow cytometry refers to the high mitochondrial membrane potential, while only FL1 positive means the low mitochondrial membrane potential.

**Figure 10 jof-08-00841-f010:**
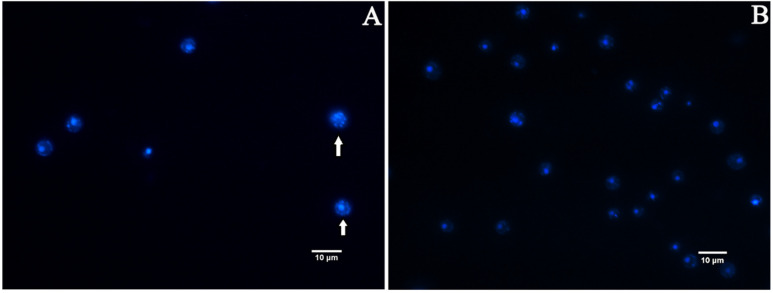
Effect of EA on nuclear fragmentation in *C. albicans* assayed by DAPI staining. (**A**) Cells exposed to 32 μg/mL EA. (**B**) Control. Nuclear fragmentations are marked by arrow.

**Table 2 jof-08-00841-t002:** The MICs and SMIC_80_s of EA, against *Candida albicans* and its biofilm.

			Irradiation	Time			Fluconazole
	0 min	5 min	10 min	20 min	40 min	60 min	
SMIC_80_ (μg/Ml)	256–512	256–512	128–256	64–128	16–32	16	>1024
MIC (μg/mL)	128	16	8	4	2	1	0.5

## Data Availability

Not applicable.

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
