# Peer review of "Inhibitory Effects and Mechanism of Action of Elsinochrome A on Candida albicans and Its Biofilm"

_jof, 2022, doi:10.3390/jof8080841_

Round 1

Reviewer 1 Report

It is an interesting and novel study showing the activity of effect of Elsinochrome A (EA) against different stage of C. albicans biofilms. However, there are multiple flaws, mainly in the presentation of results.

There are many grammar errors, and the use of acronyms is not followed.

Line 39-40 check the sentence, it is an alternative method for fungal infection?.

line 59, it is not clear whether the results obtained are from multiple strains. In the methodology, it is mentioned *Strains were*

(if multiple strains, please provide results for each strain)

Line 63:  check the sentence:

Fluconazole was purchased from Solarbio as a positive control

Line 69: explain why 10e6 cells were used if you followed CLSI guidelines (this is extremely important).

Line 73 fluconazol

 Line 75: how long?

Line 80 (all manuscript): please organise the concentrations in ascending order, e.g. 2-128

Line 84: meaning of the abbreviation

Line 95: fluconazol

Line 113:  You are not quoting a reference method, you are quoting the method of the reference 20. Please adjust the sentence. 

Line 195 meaning of the abbreviation

line 206, table 1, add a column of references (provide the references).

table 2: specify which one is biofilm

line 225: drug

Line 227 why didn't you use the abbreviation before? FLZ

Lines 238-240 Reached - over use 

Figure 2: remove min, it is duplicated 

line 247, why 90 min? (see line 240)

Lines 261-263 we, CV, And.

figure 3: if there is the same statistical difference, why use the lines (b)? Why don't you use them in the other graph (a)?

Figure 4 color conventions

a, b, c, d ?

Figure 5: What are the small dots in (8-32 ug/mL photos)?

Figure 8: check asterisks arrangement

B, y-axis title

line 343: check the sentence

Line 369 PACT

Line 370 ROS

Line 383: Why did you use fluconazole as a control?, it has only moderate antibiofilm activity and resistance is occurring. please discuss it.

Line 388: add a specific reference for fungal biofilms

Line 435 ROS

Discussion, Add a paragraph about difficulties and weaknesses of this study

Line 448, discuss how it could be used in this scenario

Reviewer 2 Report

Title: suggest to rephrase- suggestion: inhibitory effects and mechanism of action of elsinochrome A .......

Abstract - grammar and language changes are required. Suggest to improve the clarity of the abstract presentation

Introduction - suggest to provide more information on the source and properties of elsinochrome A

Materials and methods – need some information on

i.                 C. albicans strain, on its fluconazole sensitivity, biofilm producing ability

ii.               the method for preparation of elsinochrome A. What was the method to prepare purified elsinochrome A for testing and the concentration required?

iii.             Basic properties of elsinochrome A- solubility, solvent used?

iv.             Light source (for example, instrument, LED, Energy Density (Fluence) (J/cm2) Spot Size/Fiber Surface Area (cm2) etc..

v.               Did the light source affect Candida growth? Why 470 nm was selected?

Not too clear with the method described in section 2.9..

If instrument or kit were used, please include the manufacturers information  in the text, refer to section 2.11.1, 2.12

Results

Lane 214 – yeast instead of bacteria

Figure 4 – no indication of A, B, C, D in the figure

Figure 7 – top panel, the images are not clear. Is it possible to use higher magnification

Discussion

Is elsinochrome A safe for application – what will be the route of administration for therapy?

Round 2

Reviewer 1 Report

Some issues to consider

Line 41: `Photodynamic antimicrobial chemotherapy (PACT) is one of the alternative methods for traditional fungal infection treatment`. However, the reference used is about PACT in cancer. Why you said *traditional* method for fungal infection treatment? e.g., References (6-9) 2020-2022.

(innovator method?).

Line 70 meaning of the acronym YPD

Line 72 16 days?

Line 73 adequate water? Please specify

Line 73-77. Reference?

Line 84 replace the word measured by determined

155 please remove the word designed

Line 287 showed

Line 304 remove crystal violet

Figure 4. Significant differences?

figure 10 b is not clear, please replace or improve the image quality.

Line 419. Consider add a reference about candida albicans mucosal infections

e.g DOI: 10.4103/jgid.jgid_74_19

line 438. Add in a FLC-sensitive strain (MIC 0.5, SMIC80 >1024). To confirm those results in a resistant strain is recommended (please consider mentioning this).

Line 440  Full stop at the end.

Line 448 showed

Add that a weakness is that EA was only evaluated in one strain, more strains are needed mainly resistant to antifungals with different mechanisms, e.g. mutations in genes associated with efflux pumps.

add information about the strain CAMS-CCPM-D-52061 (source, date, clinical specimen?)

Reviewer 2 Report

Line 17: is it RT-qPCR (give full name)

Line 32: is it immune defence rather than homeostasis

Line 70: what is EA(98%)? Percentage shows EA quality? How the authors determine the purity of EA (based on which lab method? ). I noted that the source of EA (Shiraia bambusicola) was different from the initial report by Chen et al. This is not explained.

Line 81: Solarbio (source, country)

Line 91: FLC (concentration range?)

Line 93: which instrument was used for light irradiation experiment? Explain why the irradiation parameters were chosen? Any pilot study to show that the parameters chosen were the best?

Line 135: give citation for reference 24

Line 188: conductivity meter (brand, manufacturer)

Line 236: t-test? P value <0.05?

Line 105, 248: log instead of lg

Table 2: Please clarify which one is planktonic MIC? And which one is biofilm MIC?  Did fluconazole-treated C. albicans cultures demonstrated any increased antifungal effects upon light irradiation?

Line 254: SMIC stands for sessile MIC? Not stated. or it should be MBIC?

Line 259: Figure 1, why the starting inocula (y axis, log CFU/ml) for all EA with various concentration at 0 h were different? I thought the inocula should be standardized at 0 hour.

Line 274: EA upon light irradiation?

Figure 2: What caused C. albicans growth inhibition at 0 irradiation hour. This was not explained.

Figure 4: Was there any significant difference in the metabolic activity of C. albicans upon treatment  with  different concentrations of EA?

Line 302: scavenging – means biofilm eradication?

Figure 7. Are we expected to see some fluorescence for Fig 7A1, B1, C1, and D1. Also would like to suggest the authors to give more explanation  for their observation for Fig 7 A2-D2 in the legend. The small dots represent whole cells of C. albicans, but why there were lower number of untreated cells (A2)  and B2?  any positive control was used?

Figure 8. What is DCF, explain Figure 8A in the  legend, any positive control was used?

Figure 9: Add some explanation to Figure 9. any positive control was used?

Figure 10. How to interpret Figure 10 result on nuclear fragmentation, any positive control was used?

Line 513: any supporting evidence for nano-targeted drug delivery system

Line 520: The treatment with 1024μg/mL EA and irradiation (470nm; 90mW/cm2; 324J/cm2) had significant effects on pre-formed mature biofilms, with a clearance rate of 35.16%. The significance effects were not indicated in Figure 4

Round 3

Reviewer 2 Report

The authors have answered my queries.

qRT-PCR- Real-Time Quantitative Reverse Transcription PCR

The light emitting diode (LED) lamp (90 mW cm2; Cidly Optoelectronic Technology Co., Ltd, China), although mentioned in the authors response,  should be  also mentioned in the manuscript

Was it not necessary to have positive controls for some of the experiments?  May be this can be discussed in the limitation and future experiments
